# Synthesis of New Planar-Chiral Linked [2.2]Paracyclophanes-*N*-([2.2]-Paracyclophanylcarbamoyl)-4-([2.2]Paracyclophanylcarboxamide, [2.2]Paracyclophanyl-Substituted Triazolthiones and -Substituted Oxadiazoles

**DOI:** 10.3390/molecules25153315

**Published:** 2020-07-22

**Authors:** Ashraf A. Aly, Stefan Bräse, Alaa A. Hassan, Nasr K. Mohamed, Lamiaa E. Abd El-Haleem, Martin Nieger

**Affiliations:** 1Chemistry Department, Faculty of Science, Minia University, El-Minia 61519, Egypt; alaahassan2001@mu.edu.eg (A.A.H.); nasrmohamed603@yahoo.com (N.K.M.); lamiaaelsayed2013@yahoo.com (L.E.A.E.-H.); 2Institute of Organic Chemistry, Karlsruhe Institute of Technology, 76131 Karlsruhe, Germany; 3Institute of Biological and Chemical Systems–Functional Molecular Systems (IBCS-FMS), Karlsruhe Institute of Technology, Hermann-von-Helmholtz-Platz 1, D-76344 Eggenstein-Leopoldshafen, Germany; 4Department of Chemistry, University of Helsinki, PO Box 55 (A. I. Virtasen aukio I), 00014 Helsinki, Finland; martin.nieger@helsinki.fi

**Keywords:** HPLC, chiral *N*-([2.2]-paracyclophanylcarbamoyl)-4-([2.2] paracyclophanylcarboxamide, hydrazinecarbothioamide-paracyclophanes, paracyclophanyl-1,2,4-triazol-3-thione, paracyclophanyl)-1,3,4-oxadiazoles

## Abstract

The manuscript describes the synthesis of new racemic and chiral linked paracyclophane assigned as *N*-5-(1,4(1,4)-dibenzenacyclohexaphane-1^2^-yl)carbamoyl)-5’-(1,4(1,4)-dibenzenacyclohexaphane-1^2^-yl)carboxamide. The procedure depends upon the reaction of 5-(1,4(1,4)-dibenzenacyclohexaphane-1^2^-yl)hydrazide with 5-(1,4(1,4)-dibenzenacyclohexaphane-1^2^-yl)isocyanate. To prepare the homochiral linked paracyclophane of a compound, the enantioselectivity of 5-(1,4(1,4)-dibenzenacyclohexaphane-1^2^-yl)carbaldehyde (enantiomeric purity 60% ee), was oxidized to the corresponding acid, which on chlorination, gave the corresponding acid chloride of [2.2]paracyclophane. Following up on the same procedure applied for the preparation of racemic-carbamoyl and purified by HPLC purification, we succeeded to obtain the target *Sp*-*Sp*-*N*-5-(1,4(1,4)-dibenzenacyclohexaphane-1^2^-yl)carbamoyl)-5’-(1,4(1,4)-dibenzenacyclohexaphane-1^2^-yl)carboxamide. Subjecting *N*-5-(1,4(1,4)-dibenzenacyclohexaphane-1^2^-yl)hydrazide to various isothiocyanates, the corresponding paracyclophanyl-acylthiosemicarbazides were obtained. The latter compounds were then cyclized to a new series of 5-(1,4(1,4)-dibenzenacyclohexaphane-1^2^-yl)-2,4-dihydro-3*H*-1,2,4-triazol-3-thiones. 5-(1,4(1,4)-Dibenzenacyclohexaphane-1^2^-yl)-1,3,4-oxadiazol-2-amines were also synthesized in good yields via internal cyclization of the same paracyclophanyl-acylthiosemicarbazides. NMR, IR, and mass spectra (HRMS) were used to elucidate the structure of the obtained products. The X-ray structure analysis was also used as an unambiguous tool to elucidate the structure of the products.

## 1. Introduction

Compounds comprising the −NH–NH–C=O moiety are known as acylhydrazide linkers. More specifically, acylhydrazide-based compounds have shown antioxidant activities [1,2,3,4]. Hydrazides and carbohydrazides have been described as useful building blocks for the assembly of various heterocyclic rings [5]. A large number of aliphatic, alicyclic, aromatic and heterocyclic carbohydrazides [6,7,8,9], their derivatives, and related compounds are reported to present a plethora of biological activities [10,11,12,13,14,15,16]. 

3,4-Disubstituted-1*H*-1,2,4-triazole-5(4*H*)thiones have gained considerable importance in medicinal chemistry due to their potential anticancer [17,18,19], antimicrobial [20], antioxidant, antitumor [21], anti-tuberculosis [22], anticonvulsant [23], fungicidal [24], antiepileptic drugs [25], and anti-inflammatory activity [26]. Although they have mainly been screened for antibacterial, antifungal, anti-inflammatory, and antiproliferative activity [27,28,29,30,31], only a few studies describe their use as metalloenzyme inhibitors such as the dicopper dopamine-β-hydroxylase [32], the TNF-α converting enzyme [33], ADAMTS-5 [34], and urease [35]. A few triazolthione analogues with no amino group at the 4-position were reported to be modest inhibitors of the IMP-1 MBL [36,37] or were shown to be inactive against the CcrA, ImiS, and L1 MBLs at 50 μM [38]. Other triazolthione compounds with an alkylated sulfur atom have also been published more recently [39,40], and the structure of the complex formed by one of these compounds with VIM-2 showed that the two zinc atoms were coordinated by the nitrogen atoms at the 1- and 2-positions of the heterocycle [41,42]. 1,2,4-Triazolthione derivatives have been prepared successfully by various methods. The most common classical method is the dehydrative cyclization of different hydrazinecarbothioamides in the presence of basic media using various reagents such as sodium hydroxide [43], potassium hydroxide [44], sodium bicarbonate [45], and besides that, the acidic ionic liquid condition can be used for such cyclization followed by neutralization [46]. 

1,3,4-Oxadiazoles are an interesting class widely applied in the development of advanced electroluminescent and electron-transport materials [47,48]. In other cases, they have exhibited a variety of biological effects such as antiviral [49], antitumor [50], and anti-inflammatory [51] activities. As a design element in medicinal chemistry, 1,3,4-oxadiazoles are deployed for several purposes [52,53]. The commonly used synthetic route for 1,3,4-oxadiazoles includes reactions of acid hydrazides (or hydrazine) with acid chlorides/carboxylic acids and direct cyclization of diacylhydrazines using a variety of dehydrating agents such as phosphorous oxychloride [54], thionyl chloride [55], phosphorous pentaoxide [55], triflic anhydride [56], polyphosphoric acid [57], and a direct reaction of the acid with (*N*-isocyanimino)triphenylphosphorane [58,59,60,61].

More than six decades ago from the discovery of [2.2]paracyclophane, its derivatives have been the subject of particular interest [62,63,64,65]. Most of the unique properties of these cyclophanes are the result of the rigid framework and the short distance between the two aromatic rings within the [2.2]paracyclophane unit. The synthesis of [2.2]paracyclophane derivatives has suffered from multi-step procedures and consequently, poor yields of the desired products have been obtained [66], therefore, finding out simple methods of moderate to good yields of these compounds have been given considerable attention [66]. 

Prompted by the aforesaid properties about acylhydrazide linkers and their biological activity, in addition to the fact that planar chiral [2.2]paracyclophanes are useful synthons, from a material perspective, they can be incorporated into conjugated polymeric systems for chiroptical and optoelectronic properties. These compounds show broad applications in bio- and materials science, therefore, we decided to investigate the synthesis of homochiral linked paracyclophanes such as *N*-5-(1,4(1,4)-dibenzenacyclohexaphane-1^2^-yl)carbamoyl)-5’-(1,4(1,4)-dibenzenacyclohexaphane-1^2^-yl)carboxamide (Figure 1).

We previously reported that some paracyclophane-heterocycles such as methyl 2-(2-(4’-[2.2]paracyclophanyl)-hydrazinylidene)-3-substituted-4-oxothiazolidin-5-ylidene)acetates displayed anticancer activity against a leukemia subpanel, namely, RPMI-8226 and SR cell lines. The cytotoxic effect showed selectivity ratios ranging between 0.63 and 1.28 and between 0.58 and 5.89 at the GI50 and total growth inhibition (TGI) levels, respectively [62]. Therefore, we are aiming to prepare other classes such as triazolethione and oxadiazole moieties linked to the paracyclophane molecule. Figure 2 summarizes some of the routes utilized to prepare paracyclophanyl-triazole-3-thiones and -paracyclophanyl-2-substituted amino-1,3,4-oxadiazoles from acylhydrazinecarbothioamides [62].

## 2. Results and Discussion

The synthesis of *N*-5-(1,4(1,4)-dibenzenacyclohexaphane-1^2^-yl)carbamoyl)-5’-(1,4(1,4)-dibenzenacyclohexaphane-1^2^- yl)carboxamide (**3**) could be obtained from the reaction of *racemic*-*N*-5-(1,4(1,4)-dibenzenacyclohexaphane-1^2^-yl)hydrazide (*rac*-**1**) with *racemic-**N*-5-(1,4(1,4)-dibenzenacyclohexaphane-1^2^-yl)isothiocyanate (*rac*-**2**) (Scheme 1). 

The strategy of preparing compounds **1** and **2** was divided into two parts; firstly, starting by the parent hydrocarbon **4** as a commercial product. Compound **4** was then converted into the acid chloride derivative **6** [67], by the procedure described in Scheme 2, which consisted first of the conversion of **4** into **5** with the oxalyl chloride/aluminium trichloride. Heating of **5** in refluxing chlorobenzene caused decarbonylation to give **6**. Subsequently, the resulting acid chloride **6** was subjected toward esterification using ethanol to give compound **7** [67] (Scheme 2). Finally, the ester **7** was refluxed with a hydrazine hydrate in different solvents; however, the reaction failed to give the target α-ketohydrazine **1** in good yields. Whereas, heating **7** directly with an excess of the hydrazine hydrate afforded the corresponding *racemic*-carbohydrazide **1** in 80% yield (Scheme 2). Secondly, conversion of **6** into **2**, was achieved by the reaction of **6** with sodium azide in acetone/H_2_O to give the corresponding carbonylazide **8** [68] in 95% yield (Scheme 3). Whereas heating **8**, under Ar, in toluene afforded the second target molecule **2** [68] in 70% yield (Scheme 3). 

The structure of newly prepared compound **1** was proved by NMR spectra. The ^1^H-NMR spectrum revealed two singlets at δ = 9.09 and 4.46 assigned to NH and NH_2_ protons. The ^13^C-NMR spectrum showed the carbonyl-carbon at δ = 167.8 (C=O), whereas the four distinctive CH_2_-bridged carbons resonated at δ = 34.8 (CH_2_-1’), 34.7 (CH_2_-10’), 34.5 (CH_2_-9’), and 34.2 (CH_2_-2’). The IR spectrum revealed the absorption of NH_2_, NH, and the carbonyl groups at *ṽ* = 3352–3214 NH_2_, 3196, and 1632, respectively. The X-ray structure analysis was used to elucidate the structure feature of compound **1** as shown in Figure 3.

### 2.1. Preparation of Compound ***3***

As above mentioned in Scheme 1, the diastereomer **3** was obtained via the reaction of carbohydrazide paracyclophane (**1**) and paracyclophane isocyanate (**2**) in a mixture of absolute EtOH:DMF (i.e., 25:1 by volume in mL). Several trials including different solvents such as EtOH, EtOH/Et_3_N, DMF, Toluene/Et_3_N, and propanol failed to give good yields. However, a mixture of EtOH to DMF (25:1) gave compound **3** in 70% yield.

The strategy of preparing *Sp*-*Sp*-**3** was started by preparation of *Sp*-4-formyl([2.2]paracyclophane (**9**) (60% ee) [68]. To prove the enantiomeric purity of **9**, a chiral HPLC analysis was conducted, it was found that **9** has an enantiomeric excess of 60% (Figure 4), meaning it is not completely *Sp*-pure. Oxidation of **9** gave the target acid **10** [68], which on chlorination via reaction with thionyl dichloride/DMF gave **6** [67] (Scheme 4). Subsequently, repeating the previous steps in Scheme 1, Scheme 2 and Scheme 3, compounds **1–3** were prepared in their ralemic forms. Applying the HPLC separation on **3**, the desired pure chiral (*Sp*-*Sp*)-*N*-([2.2]-paracyclophanylcarbamoyl)-4-([2.2]paracyclophanylamide (**3**) (Figure 5) was obtained. 

The structure of the compounds diastereomer-**3** and/or pure chiral *Sp*-*Sp*-**3** was proved by the NMR spectroscopic analysis. It is clearly apparent that the same number of protons and carbon signals was assigned for both aforesaid compounds. The twenty-four paracyclophanyl (PC)-aromatic carbons and eight PC-CH_2_-bridged carbons in addition to the two carbonyl carbons appeared in the ^13^C-NMR spectrum of pure chiral *Sp*-*Sp*-**3**, since the two paracyclophanyl moieties were electronically different via attachment with two different functional groups.

### 2.2. Synthesis of Triazolethiones ***12a****–****f***

The synthesis of these nitrogen-containing heterophanes led to the idea that the heterophane 5-(1,4(1,4)-dibenzenacyclohexaphane-1^2^-yl)-4-substituted-2,4-dihydro-3*H*-1,2,4-triazol-3-thiones **12a**–**f** in 72–78% yields could be obtained from the cyclization of **11a**–**f** [62] in an alkaline medium (Scheme 5). Compounds **11a**–**f** were previously prepared by the reaction of compound **1** with isothiocyanates (Scheme 5) [62]. All compounds of the series **12a**–**f** provided analytical data in full agreement with the desired structures (see Experimental and Appendix A). The structures of compounds **12a** and **12d** were completely proved by the X-ray structure analyses, as shown in Figure 6 and Figure 7, respectively.

### 2.3. Conversion of N-Substituted-5-(1,4(1,4)-Dibenzenacyclohexaphane-1^2^-yl)Hydrazinecarbothioamides ***11a**–**f*** into 5-(1,4(1,4)-Dibenzenacyclohexaphane-1^2^-yl)-N-Substituted-1,3,4-Oxadiazol-2-Amines ***13a**–**e***

The one-pot synthesis including gentle heating of **1a**–**f** in tetrahydrofuran (THF) together with 0.5 mL of Et_3_N afforded directly the corresponding 5-(1,4(1,4)-dibenzenacyclohexaphane-1^2^-yl)-*N*-substituted-1,3,4-oxadiazol-2-amines **13a**–**f** in 63–68% yields (Scheme 6). Compound **13a** was identified from spectroscopic data as 5-(1,4(1,4)-dibenzenacyclohexaphane-1^2^-yl)-1,3,4-oxadiazol-2-amine. As an example, the ^1^H-NMR spectrum showed two broad singlets at *δ* = 10.04 for NH and CH-5-PC. The ^13^C-NMR spectrum revealed the oxadiazole-C-2 and oxadiazole carbons at *δ* = 162.7 for-C-2 and *δ* = 160.0 for C-5. The four PC-CH_2_ carbons appeared at *δ* = 36.2, 36.1, 36.0, and 35.9. Elemental and mass spectroscopy indicated the molecular formula of **13a** as C_24_H_21_N_3_O. The structure of **13a** was proved by the X-ray structure analysis as shown in Figure 8. The X-ray was used to prove the structure of compounds **13a** and **13e**, as shown in Figure 8 and Figure 9, respectively.

## 3. Experimental

### 3.1. Material and Methods

The IR spectra were recorded by the ATR technique (ATR (Attenuated Total Reflection)) with an FT device (FT-IR Bruker IFS 88, Bremen, Germany), Institute of Organic Chemistry, Karlsruhe University, Karlsruhe, Germany. The NMR spectra (Appendix A) were measured in DMSO-*d*_6_ on a Bruker AV-400 spectrometer (Germany), 400 MHz for ^1^H, and 100 MHz for ^13^C; and the chemical shifts are expressed in δ (ppm), versus internal tetramethylsilane (TMS) = 0 for ^1^H and ^13^C, and external liquid ammonia = 0. The description of signals includes: s (singlet), d (doublet), t (triplet), q (quartet), m (multiplet), dd (doublet of doublet), ddd (doublet of dd), dt (doublet of triplet), td (triplet of doublet), bs (broad singlet), and m (multiplet). Mass spectra were recorded on a FAB (fast atom bombardment) Thermo Finnigan Mat 95 (70 eV) (Thermo Electron (Bremen) GmbH, Barkhausenstr. 2 D-28197 Bremen). For the high-resolution mass, the following abbreviations were used: Calc.: Theoretical calculated mass; found: Mass found in the analysis, Institute of Organic Chemistry, Karlsruhe Institute of Technology, Karlsruhe, Germany. The TLC was performed on analytical Merck 9385 silica aluminium sheets (Kieselgel 60) with a Pf_254_ indicator; the TLCs were viewed at λ_max_ = 254 nm, crude products were purified by flash chromatography with Silica gel 60 (0.040 × 0.063 mm, Geduran) (Merck, Germany).

Compounds **2** and **5**–**10** were prepared according to the literature [67,68]. Compounds **11a**–**f** were prepared according to the methodology mentioned in reference [62].

### 3.2. Racemic-N-5-(1,4(1,4)-Dibenzenacyclohexaphane-1^2^-yl)hydrazide *(**1**)*

Under an argon atmosphere, a mixture of ethyl [2.2]paracyclophane-4-carboxylate (**8**) [39] (5.00 g, 18.0 mmol, 1.00 equiv.) was dissolved in 25 mL of hydrazine monohydrate and heated under reflux for 14 h. The reaction mixture was then cooled to room temperature until a precipitate was formed (24 h). The precipitate was then filtered and washed with 150 mL of water (three times) followed by 100 mL of heptane and then dried. The white product of *racemic*- or *Sp*-**1** was obtained and recrystallized from ethanol.

*Racemic-**N*-5-(1,4(1,4)-dibenzenacyclohexaphane-1^2^-yl)hydrazide (**1**). Colorless crystals (EtOH), yield 3.80 g, (80%), m.p. 230–232 °C, ^1^H-NMR (400 MHz, DMSO-*d*_6_, ppm) δ = 9.09 (s, 1H, NH), 6.66 (d, *J* = 1.9 Hz, 1H, PC-H), 6.60 (d, *J* = 7.7 Hz, 1H, PC-H), 6.57 (dd, *J* = 7.7, 1.9 Hz, 1H, PC-H), 6.53 (d, *J* = 1.3 Hz, 2H, PC-H), 6.47 (d, *J* = 7.7 Hz, 1H, PC-H), 6.42 (d, *J* = 7.8 Hz, 1H, PC-H), 4.46 (brs, 2H, NH_2_), 3.08 (dddd, *J* = 12.7, 9.9, 6.4, 3.1 Hz, 3H, CH_2_-CH_2_), 3.04–2.85 (m, 4H, CH_2_-CH_2_), 2.81 (ddd, *J* = 12.5, 9.5, 6.3 Hz, 1H, CH_2_-CH_2_). ^13^C-NMR (100 MHz, DMSO-*d*_6_, ppm) δ = 167.8 (C=O), 139.3 (PC-C-6’), 139.3 (PC-C-11’), 139.0 (PC-C-14’), 138.8 (PC-C-3’), 135.5, 134.5, 133.7, 132.5, 132.4, 131.6, 131.43 (PC-CH), 131.4 (PC-C-4’), 34.8 (PC-CH_2_-1’), 34.7 (PC-CH_2_-10’), 34.5 (PC-C CH_2_-9’), 34.2 (PC-C CH_2_-2’). IR (ATR, cm^-1^) *ṽ* = 3352–3214 (br, NH_2_), 3196 (w, NH), 2927 (s, Ar-CH), 2848 (m, aliph-CH), 1632 (s, CO). MS (FAB) *m/z* (%) = 266.3 [M]^+^ (100). HRMS (EI, [M]^+^, C_17_H_18_O_1_N_2_) calc.: 266.1419, found: 266.1418.

### 3.3. Preparation of Compound Diasteromer-***3*** or Sp-Sp-***3***

In a 100 mL round-bottomed flask, a mixture of carbohydrazide paracyclophane (**1**, 160 mg, 601 μmol, 1.00 equiv.) and paracyclophane isocyanate (**2**, 150 mg, 601 μmol, 1.00 equiv.) in a mixture of absolute EtOH: DMF (25:1 by volume in mL) was heated in an oil bath at 70 °C for 4 h. The formed precipitate was filtered and washed with heptane several times (3 × 20 mL). 

Diastereomeric Mixture of N-5-(1,4(1,4)-Dibenzenacyclohexaphane-1^2^-yl)carbamoyl)-5’-(1,4(1,4)-dibenzenacyclohexaphane-1^2^-yl)carboxamide (**3**). Colorless crystals (EtOH), yield 0.36 g (70%), m.p. 310–2 °C, ^1^H-NMR (400 MHz, DMSO-d_6_, ppm) δ = 9.72 (d, J = 2.4 Hz, 1H, NH^1^-hydrazide), 8.39 (dd, 1H, NH^2^-hydrazide), 7.99 (s, 1H, NH-amide), 6.93–6.87 (m, 2H, PC-H), 6.76–6.71 (m, 2H, PC-H), 6.65–6.63 (m, 1H, PC-H), 6.55–6.48 (m, 5H, PC-H), 6.44–6.31 (m, 4H, PC-H), 3.79 (ddd, J = 12.6, 9.0, 3.4 Hz, 1H, PC-CH_2_-CH_2_), 3.17–3.09 (m, 2H, PC-CH_2_-CH_2_), 3.09–2.99 (m, 4H, PC-CH_2_-CH_2_), 2.99–2.90 (m, 6H, PC-CH_2_-CH_2_), 2.90–2.67 (m, 3H, PC-CH_2_-CH_2_). ^13^C-NMR (100 MHz, DMSO-d_6_, ppm) δ = 168.8 (C=O-hydrazide), 156.3 (C=O-amide), 140.7, 140.3 (PC-C-6’,6”), 140.2, 140.1 (PC-C-11’,11”), 139.9, 139.7 (PC-C-14’,14”), 139.6, 139.4 (PC-C-3’,3”), 139.1, 138.1, 136.4, 135.4, 133.5, 133.3, 133.2, 133.1, 132.9, 132.8, 132.3, 132.2, 131.9 (PC-CH), 128.8, 127.7 (PC-C-4’), 126.2 (PC-CH), 35.5 (PC-CH_2_), 35.2 (PC-2CH_2_), 34.9 (PC-2CH_2_), 34.7, 33.5, 33.2 (PC-CH_2_). IR (ATR, cm^−1^) ṽ = 3428–3224 (br, NH), 3104–3087 (w, NH), 2955–2893 (w, NH), 2864 (w, Ar-CH), 2853 (w, aliph-CH), 1642, 1572 (s, CO). MS (FAB) m/z (%) = 516.3 [M + H]^+^ (70). HRMS (EI, [M + H]^+^, C_34_H_34_O_2_N_3_) calc.: 516.2651, found: 516.2652.

*(Sp-Sp)-N-5-(1,4(1,4)-Dibenzenacyclohexaphane-1^2^-yl)carbamoyl)-5’-(1,4(1,4)-dibenzenacyclohexaphane-1^2^-yl)carboxamide* (**3**). Colorless crystals (EtOH), yield 0.26 g (50%), m.p. 310–2 °C, [α]_D_= + 41.8 (c 0.004, CH_2_Cl_2_). ^1^H-NMR (400 MHz, DMSO-*d*_6_, ppm) δ = 9.73, (s, 1H, NH^1^-hydrazide), 8.40 (s, 1H, NH^2^-hydrazide), 8.00 (s, 1H, NH-amide), 7.07–6.85 (m, 2H, PC-H), 6.84–6.74 (m, 2H, PC-H), 6.65 (dd, *J* = 7.7, 1.8 Hz, 1H, PC-H), 6.58–6.45 (m, 6H, PC-H), 6.40 (d, *J* = 0.7 Hz, 2H, PC-CH), 6.33 (dd, *J* = 7.8, 1.8 Hz, 1H, PC-H), 3.84–3.77 (m, 1H, PC-CH_2_-CH_2_), 3.16–3.08 (m, 2H, PC-CH_2_-CH_2_), 3.06–2.91 (m, 10H, PC-CH_2_-CH_2_), 2.89–2.69 (m, 3H, PC-CH_2_-CH_2_). ^13^C-NMR (100 MHz, DMSO-*d*_6_, ppm) δ = 168.6 (C=O-hydrazide), 156.2 (C=O-amide), 140.6, 140.2 (PC-C-6’,6”), 139.9 (2C-PC-C-11’,11”), 139.5, 139.4 (PC-C-14’,14”), 139.0, 138.5 (PC-C-3’,3”), 136.2, 135.5, 135.3, 133.6 (PC-CH), 133.5 (PC-2CH), 133.1 (PC-CH), 132.9 (PC-2CH), 132.4, 132.3, 132.1 (PC-CH), 128.8, 127.2 (PC-C-4’,4’’), 125.9 (PC-2CH), 35.3, 35.2, 35.1, 35.0, 34.9, 33.5, 33.4, 33.2 (PC-CH_2_). IR (ATR, cm^-1^) *ṽ* = 3342–3275 (br, NH), 3197–3012 (w, NH), 2962–2893 (w, NH), 2856 (w, Ar-CH), 1643, 1572 (s, CO). MS (FAB) *m/z* (%) = 516.3 [M + H]^+^ (30). HRMS (EI, [M + H]^+^, C_34_H_34_O_2_N_3_) calc.: 516.2651, found: 516.2652.

### 3.4. High-Performance Liquid Chromatography (HPLC)

Purification of *N*-5-(1,4(1,4)-dibenzenacyclohexaphane-1^2^-yl)carbamoyl)-5’-(1,4(1,4)-dibenzenacyclohexaphane-1^2^-yl)carboxamide (*Sp*-*Sp*-**3**) (60% ee) were conducted using preparative HPLC setups: The JASCO HPLC System (LC-NetII/ADC) (JASCO, Inc., Pfungstadt, Germany) equipped with two PU-2087 Plus pumps, a CO-2060 Plus thermostat, an MD-2010 Plus diode array detector, and a CHF-122SC fraction collector of ADVANTEC (München, Germany). For the purification, a Daicel Chiralpak (AZ-H 20 × 250 mm, particle size of 5 µm) (Daicel Chiralpak, Tokyo, Japan) was used with the HPLC-grade acetonitrile as a mobile phase. Detection was conducted at 256 nm.

Analysis of the enantiomeric excess was conducted using an AGILENT HPLC 1100 series system with a G1322A degasser, a G1211A pump, a G1313A autosampler, a G1316A column oven, and a G1315B diode array system (Agilent, Waldbronn, Germany). Chiralpak OD-H (4.6 × 250 mm, 5 µm particle size) columns (Agilent, Waldbronn, Germany) were used with the HPLC-grade *n*-hexane/isopropanol as a mobile phase. The y-axis of the chromatogram is a measure of the intensity of absorbance (in units of mAU, or milli-Absorbance Units). The x-axis is in units of time (typically minutes), and is used to determine the retention time (tR) for each peak.

### 3.5. Preparation of 5-(1,4(1,4)-Dibenzenacyclohexaphane-1^2^-yl)-2,4-dihydro-3H-1,2,4-triazol-3-thiones ***12a**–**f***

A stirring mixture of hydrazinecarbothioamide derivatives **11a**–**f** (1 mmol) and 100 mL of sodium hydroxide (1 mmol, as a 2N solution) was refluxed for 2–4 h. After cooling, the solution was acidified with 100 mL of hydrochloric acid (6M) and the formed precipitate was filtered. The precipitate was then recrystallized from ethanol. 

*5-(1,4(1,4)-Dibenzenacyclohexaphane-1^2^-yl)-4-phenyl-2,4-dihydro-3H-1,2,4-triazol-3-thione* (**12a**). Colorless crystals (DMSO), 300 mg (78%), m.p. 150–2 °C, *R_f_* = 0.5 (Hexane: Ethyl acetate; 10:1). ^1^H-NMR (400 MHz, DMSO-*d*_6_, ppm) δ = 14.20 (s, 1H, NH), 7.31–7.27 (m, 3H, Ph-H), 7.07–7.03 (m, 2H, Ph-H), 673–6.69 (m, 1H, PC-H), 6.63 (d, *J* = 2.0 Hz, 1H, PC-H), 6.58–6.49 (m, 3H, PC-H), 6.36–6.30 (m, 2H, PC-H), 3.10–2.89 (m, 6H, PC-CH_2_), 2.86–2.77 (m, 2H, PC-CH_2_). ^13^C-NMR (100 MHz, DMSO-*d**_6_*, ppm) δ = 167.8 (C=S), 151.7 (triazole-C5), 139.7 (PC-C-6’), 139.2 (PC-C-11’), 139.1 (PC-C-14’), 139.0 (PC-C-3’), 135.7 (Ph-C), 135.0, 134.1, 133.5 (PC-CH), 132.9 (PC-2CH), 132.0, 131.0 (PC-CH), 128.8 (PC-C-4’), 128.6, 128.4 (Ph-2CH), 125.3 (Ph-CH), 34.8 (PC-CH_2_-1’), 34.7 (PC-CH_2_-10’), 34.4 (PC-C H_2_-9’), 33.6 (PC-CH_2_-2’). IR (ATR, cm^−1^) *ṽ* = 3099 (m, Ar-CH), 2924 (s, aliph-CH), 1499 (s, C-S), 1333 (s, C-N). MS (FAB) *m/z* (%) = 384.1 [M + H]^+^ (100). HRMS (FAB, [M + H]^+^, C_24_H_22_N_3_^32^S_1_) calc.: 384.1534, found: 384.1526.

*5-(1,4(1,4)-Dibenzenacyclohexaphane-1^2^-yl)-4-(pyridine-3-yl)-2,4-dihydro-3H-1,2,4-triazol-3-thione* (**12b**). Colorless crystals (DMSO), 290 mg (76%), m.p. 170–2 °C, *R_f_* = 0.4 (Hexane: Ethyl acetate; 10:1). ^1^H-NMR (400 MHz, DMSO-*d*_6_, ppm) δ = 14.16 (s, 1H, NH), 8.55–8.28 (m, 2H, Pyr-H), 7.71–7.17 (m, 2H, Pyr-H), 6.96–6.94 (m, 1H, PC-H), 6.74–6.65 (m, 2H, PC-H), 6.59–6.33 (m, 4H, PC-H), 3.16– 2.91 (m, 6H, PC-CH_2_), 2.88–2.78 (m, 2H, PC-CH_2_). ^13^C-NMR (100 MHz, DMSO-*d**_6_*, ppm) δ = 167.9 (C=S), 151.4 (triazole-C5), 149.3, 148.7 (Pyr-CH), 140.0 (PC-C-6’), 139.5 (PC-C-11’), 139.2 (PC-C-14’), 139.1 (PC-C-3’), 136.5 (Pyr-C), 135.9, 135.1, 133.6, 133.1, 132.5, 132.0, 131.1 (PC-CH), 130.9 (PC-C-4’), 124.9, 123.6 (Pyr-CH), 34.7 (PC-CH_2_-1’), 34.6 (PC-CH_2_-10’), 34.4 (PC-CH_2_-9’), 33.4 (PC-CH_2_-2’). IR (ATR, cm^−1^) *ṽ* = 2924 (s, Ar-CH), 2850 (m, aliph-CH), 1483 (s, C-S), 1313 (s, C-N). MS (FAB) *m/z* (%) = 385.2 [M + H]^+^ (60). HRMS (FAB, [M + H]^+^, C_23_H_21_N_4_^32^S_1_) calc.: 385.1487, found: 385.1488.

*4-Allyl-5-(1,4(1,4)-dibenzenacyclohexaphane-1^2^-yl)-2,4-dihydro-3H-1,2,4-triazol-3-thione* (**12c**). Colorless crystals (DMSO), 270 mg (78%), m.p. 142–4 °C, *R_f_* = 0.6 (Hexane: Ethyl acetate; 10:1). ^1^H-NMR (400 MHz, DMSO-*d*_6_, ppm) δ = 14.05 (s, 1H, NH), 6.75 (dd, *J* = 7.8, 1.9 Hz, 1H, PC-H), 6.69–6.58 (m, 5H, PC-H), 6.38 (dd, *J* = 7.9, 1.8 Hz, 1H, PC-H), 5.55–5.48 (m, 1H, allyl-CH=), 4.90–4.52 (m, 2H, allyl-CH_2_=), 4.51–4.20 (m, 2H, allyl-CH_2_), 3.13–2.90 (m, 7H, PC-CH_2_), 2.82–2.77 (m, 1H, PC-CH_2_). ^13^C-NMR (100 MHz, DMSO-*d**_6_*, ppm) δ = 166.9 (C=S), 151.6 (triazole-C5), 140.4 (PC-C-6’), 139.4 (PC-C-11’), 139.0 (PC-C-14’), 138.9 (PC-C-3’), 135.9 (allyl-CH=), 135.5 (PC-C-4’), 133.5, 133.1, 133.0, 132.3, 131.5, 131.2, 125.3 (PC-CH), 117.5 (allyl-CH_2_=), 45.4 (allyl-CH_2_), 34.9 (PC-CH_2_-1’), 34.8 (PC-CH_2_-10’), 34.5 (PC-CH_2_-9’), 33.5 (PC-CH_2_-2’). IR (ATR, cm^−1^) *ṽ* = 3187 (m, Ar-CH), 2929 (s, aliph-CH), 1435 (s, C-S), 1268 (s, C-N). MS (FAB) *m/z* (%) = 348.2 [M + H]^+^ (65). HRMS (FAB, [M + H]^+^, C_21_H_22_N_3_^32^S_1_) calc.: 347,1456, found: 347,1457.

*5-(1,4(1,4)-Dibenzenacyclohexaphane-1^2^-yl)-4-ethyl-2,4-dihydro-3H-1,2,4-triazol-3-thione* (**12d**). Colorless crystals (DMSO), 240 mg (72%), m.p. 138–40 °C, *R_f_* = 0.65 (Hexane: Ethyl acetate; 10:1). ^1^H-NMR (400 MHz, DMSO-*d*_6_, ppm) δ = 13.98 (s, 1H, NH), 6.74 (dd, *J* = 7.8, 1.8 Hz, 1H, PC-H), 6.71–6.57 (m, 5H, PC-H), 6.38 (dd, *J* = 7.9, 1.8 Hz, 1H, PC-H), 3.86–3.58 (m, 2H, ethyl-CH_2_), 3.12–2.97 (m, 5H, PC-CH_2_), 2.96–2.91 (m, 2H, PC-CH_2_), 2.79–2.72 (m, 1H, PC-CH_2_), 0.81 (t, *J* = 7.1 Hz, 3H, ethyl-CH_3_). ^13^C-NMR (100 MHz, DMSO-*d**_6_*, ppm) δ = 166.1 (C=S), 151.1 (triazole-C5), 140.4 (PC-C-6’), 139.2 (PC-C-11’), 138.7 (PC-C-14’), 138.6 (PC-C-3’), 135.8, 135.3, 133.2 (PC-CH), 132.8 (PC-2CH), 132.1 (PC-CH), 131.3 (PC-CH), 125.0 (PC-C-4’), 38.4 (ethyl-CH_2_), 34.7 (PC-CH_2_-1’), 34.6 (PC-CH_2_-10’), 34.3 (PC-CH_2_-9’), 33.1 (PC-CH_2_-2’), 13.0 (ethyl-CH_3_). IR (ATR, cm^−1^) *ṽ* = 3122 (m, Ar-CH), 2932 (s, aliph-CH), 1500 (s, C-S), 1283 (s, C-N). MS (FAB) *m/z* (%) = 336.2 [M + H]^+^ (95). HRMS (FAB, [M + H]^+^, C_20_H_22_N_3_^32^S_1_) calc.: 336.1534, found: 336.1534.

*4-Cyclopropyl-5-(1,4(1,4)-dibenzenacyclohexaphane-1^2^-yl)-2,4-dihydro-3H-1,2,4-triazol-3-thione* (**12e**). Colorless crystals (DMSO), 250 mg (72%), m.p. 167–9 °C, *R_f_* = 0.65 (Hexane: Ethyl acetate; 10:1). ^1^H-NMR (400 MHz, DMSO-*d*_6_, ppm) δ = 14.27 (s, 1H, NH), 6.75–6.70 (m, 1H, PC-H), 6.67 (d, *J* = 1.8 Hz, 1H, PC-H), 6.62–6.55 (m, 4H, PC-H), 6.36 (dd, *J* = 7.8, 1.7 Hz, 1H, PC-H), 3.13–2.98 (m, 8H, PC-CH_2_), 2.91–2.86 (m, 1H, cyclopropyl-CH), 0.79–0.52 (m, 2H, cyclopropyl-CH_2_), 0.45–0.05 (m, 2H, cyclopropyl-CH_2_). ^13^C-NMR (100 MHZ, DMSO-*d*_6_, ppm) δ = 167.6 (C=S), 152.6 (triazole-C5), 139.3 (PC-C-6’), 139.2 (PC-C-11’), 139.0 (PC-C-14’), 138.9 (PC-C-3’), 134.7, 134.6, 133.0, 132.9, 132.8, 132.0, 131.0 (PC-CH), 126.8 (PC-C-4’), 34.8 (PC-CH_2_-1’), 34.7 (PC-CH_2_-10’), 34.5 (PC-CH_2_-9’), 33.4 (PC-CH_2_-2’), 25.9 (cyclopropyl-CH), 8.3 (cyclopropyl-CH_2_), 8.2 (cyclopropyl-CH_2_). IR (ATR, cm^−1^) *ṽ* = 3091 (m, Ar-CH), 2922 (s, aliph-CH), 1429 (s, C-S), 1362 (s, C-N). MS (FAB) *m/z* (%) = 348.2 [M + H]^+^ (100). HRMS (FAB, [M + H]^+^, C_22_H_22_N_3_^32^S_1_) calc.: 348.1534, found: 348.1599.

*4-Benzyl-5-(1,4(1,4)-dibenzenacyclohexaphane-1^2^-yl)-2,4-dihydro-3H-1,2,4-triazol-3-thione* (**12f**). Colorless crystals (DMSO), 310 mg (78%), m.p. 185–7 °C, *R_f_* = 0.4 (Hexane: Ethyl acetate; 10:1). ^1^H-NMR (400 MHz, DMSO-*d*_6_, ppm) δ = 14.13 (s, 1H, NH), 7.12–7.07 (m, 3H, Ph-H), 6.78–6.76 (m, 2H, Ph-H), 6.74–6.65 (m, 2H, PC-H), 6.61–6.54 (m, 3H, PC-H), 6.30–6.36 (m, 2H, PC-H), 4.98 (s, 2H, Benzyl-CH_2_), 3.06–2.89 (m, 6H, PC-CH_2_), 2.73–2.66 (m, 2H, PC-CH_2_). ^13^C-NMR (100 MHz, DMSO-*d**_6_*, ppm) δ = 167.4 (C=S), 151.8 (triazole-C5), 140.4 (PC-C-6’), 140.2 (PC-C-11’), 139.4 (PC-C-14’), 139.1 (PC-C-3’), 137.2 (Ph-C), 136.1 (PC-2CH), 135.7, 135.5 (PC-CH), 133.5 (PC-2CH), 133.1 (PC-CH), 133.0, 128.8 (Ph-CH), 128.6 (Ph-2CH), 128.4 (PC-C-4’), 125.3 (Ph-CH), 46.4 (Benzyl-CH_2_), 35.5 (PC-CH_2_-1’), 34.9 (PC-CH_2_-10’), 34.6 (PC-C H_2_-9’), 33.3 (PC-C H_2_-2’). IR (ATR, cm^−1^) *ṽ* = 3054 (w, Ar-CH), 2925 (w, aliph-CH), 1510 (w, C-S), 1183 (w, C-N). MS (FAB) *m/z* (%) = 398.2 [M + H]^+^ (100). HRMS (FAB, [M + H]^+^, C_25_H_24_N_3_^32^S_1_) calc.: 398.1691, found: 398.1692.

### 3.6. Preparation of N-Substituted 5-(1,4(1,4)-Dibenzenacyclohexaphane-12-yl)-1,3,4-Oxadiazol-2-Amines ***13a**–**e***

A stirring mixture of hydrazinecarbothioamide derivatives **11a**–**f** (1 mmol) in 100 ml tetrahydrofuran (THF) together with 0.5 mL of Et_3_N was refluxed for 12–24 h (the reaction was monitored by thin-layer chromatography). After removal of the solvent on vacuum, the crude residue was purified by column chromatography (cyclohexane/ethyl acetate 10:5) as an eluent to afford compounds **3a**–**e**. 

*5-(1,4(1,4)-Dibenzenacyclohexaphane-12-yl)-N-phenyl-1,3,4-oxadiazol-2-amine* (**13a**). Yellow crystals (Acetonitrile), 250 mg (68%), m.p. 192–4 °C, *R_f_* = 0.5 (Hexane: Ethyl acetate; 5:1). ^1^H-NMR (400 MHz, Acetone-*d*_6_, ppm) δ = 10.03 (s, 1H, NH), 7.80–7.77 (m, 1H, Ph-H), 7.48–7.38 (m, 2H, Ph-H), 7.10–7.03 (m, 2H, Ph-H), 6.99–6.96 (m, 1H, PC-H), 6.92–6.73 (m, 2H, PC-H), 6.70–6.45 (m, 4H, PC-H), 3.23–3.13 (m, 3H, PC-CH_2_), 3.12–3.01 (m, 4H, PC-CH_2_), 3.00–2.94 (m, 1H, PC-CH_2_). ^13^C-NMR (100 MHz, Acetone-*d*_6_, ppm) δ = 162.7 (oxadiazole-C2), 160.0 (oxadiazole-C5), 141.3 (PC-C-6’), 141.1 (PC-C-11’), 140.5 (PC-C-14’), 140.2 (PC-C-3’), 139.6 (Ph-C), 137.1, 135.4, 134.0, 133.9, 133.6, 133.0, 132.9 (PC-CH), 131.3, 130.3, 129.9 (Ph-CH), 125.8 (PC-C-4’), 122.8, 121.6, (Ph-CH), 35.9 (PC-CH_2_-1’), 35.7 (PC-CH_2_-10’), 35.5 (PC-CH_2_-9’), 34.7 (PC-CH_2_-2’). IR (ATR, cm^−1^) *ṽ* = 3378 (m, NH), 2927 (m, Ar-CH), 2850 (s, aliph-CH), 1587 (C=N), 1482 (Ar-C=C), 1044 (C-O-C). MS (FAB) *m/z* (%) = 368.3 [M + H]^+^ (75). HRMS (FAB, [M + H]^+^, C_24_H_22_O_1_N_3_) calc.: 368.1763, found: 368.1761.

*5-(1,4(1,4)-Dibenzenacyclohexaphane-12-yl)-N-(pyridin-4-yl)-1,3,4-oxadiazol-2-amine* (**13b**). Yellow crystals (Acetonitrile), 240 mg (66%), m.p. 212–4 °C, *R_f_* = 0.3 (Hexane: Ethyl acetate; 5:1). ^1^H-NMR (400 MHz, CDCl_3_-*d*, ppm) δ = 9.06 (br, 1H, NH), 8.68 (d, *J* = 6.1 Hz, 2H, Pyr-H), 7.75–7.72 (m, 1H, Pyr-H), 7.28–7.26 (m, 1H, Pyr-H), 6.99–6.93 (m, 2H, PC-H), 6.90–6.73 (m, 5H, PC-H), 3.21–3.12 (m, 4H, PC-CH_2_), 3.10–3.01 (m, 3H, PC-CH_2_), 2.99–2.93 (m, 1H, PC-CH_2_). ^13^C-NMR (100 MHz, CDCl_3_-*d*, ppm) δ = 167.1 (oxadiazole-C2), 159.9 (oxadiazole-C5), 140.4 (pyr-2CH), 139.9 (PC-C-6’), 139.8 (PC-C-11’), 139.3 (PC-C-14’), 138.7 (PC-C-3’), 137.2 (Pyr-C), 136.3, 135.1, 133.1 (PC-CH), 132.5, 132.2, 130.9 (PC-CH), 130.6 (Pyr-CH), 130.0 (PC-CH), 125.0 (PC-C-4’), 124.2 (Pyr-CH), 35.5 (PC-CH_2_-1’), 35.3 (PC-CH_2_-10’), 35.1 (PC-CH_2_-9’), 34.3 (PC-CH_2_-2’). IR (ATR, cm^−1^) *ṽ* = 3017 (m, NH), 2922 (s, Ar-CH), 2850 (s, aliph-CH), 1581 (s, C=N), 1548 (s, Ar-C=C), 1043 (s, C-O-C). MS (FAB) *m/z* (%) = 369.2 [M + H]^+^ (100). HRMS (FAB, [M + H]^+^, C_23_H_21_O_1_N_4_) calc.: 369.1715, found: 369.1714.

*N-Allyl-5-(1,4(1,4)-dibenzenacyclohexaphane-12-yl)-1,3,4-oxadiazol-2-amine* (**13c**). Yellow crystals (Acetonitrile), 220 mg (66%), m.p. 206–8 °C, *R_f_* = 0.7 (Hexane: Ethyl acetate; 5:1). ^1^H-NMR (400 MHz, Acetone-*d*_6_, ppm) δ = 6.91 (s, 1H, NH), 6.70–6.55 (m, 5H, PC-H), 6.44–6.37 (m, 2H, PC-H), 6.10–6.02 (m, 1H, allyl-CH=), 5.38–5.17 (m, 2H, allyl-CH_2_=), 4.07–4.05 (m, 2H, allyl-CH_2_), 3.18–3.10 (m, 3H, PC-CH_2_), 3.08–2.96 (m, 4H, PC-CH_2_), 2.95–2.89 (m, 1H, PC-CH_2_). ^13^C-NMR (100 MHz, Acetone-*d*_6_, ppm) δ = 164.3 (oxadiazole-C2), 160.0 (oxadiazole-C5), 141.3 (PC-C-6’), 140.6 (PC-C-11’), 140.3 (PC-C-14’), 139.8 (PC-C-3’), 137.1, 135.7, 135.1, 134.2 (PC-CH), 134.1 (allyl-CH=), 133.1, 132.9, 131.3 (PC-CH), 126.5 (PC-C-4’), 116.5 (allyl-CH_2_=), 46.4 (allyl-CH_2_), 36.1 (PC-CH_2_-1’), 35.8 (PC-CH_2_-10’), 35.6 (PC-CH_2_-9’), 34.8 (PC-CH_2_-2’). IR (ATR, cm^−1^) *ṽ* = 3165 (w, NH), 2929 (s, Ar-CH), 2861 (m, aliph-CH), 1557 (s, C=N), 1442 (s, Ar-C=C), 1035 (s, C-O-C). MS (FAB) *m/z* (%) = 332.2 [M + H]^+^ (100). HRMS (FAB, [M + H]^+^, C_21_H_22_O_1_N_3_) calc.: 332.1763, found: 332.1764.

*5-(1,4(1,4)-Dibenzenacyclohexaphane-12-yl)-N-ethyl-1,3,4-oxadiazol-2-amine* (**13d**). Yellow crystals (Acetonitrile), 190 mg (60%), m.p. 211–4 °C, *R_f_* = 0.7 (Hexane: Ethyl acetate; 5:1). ^1^H-NMR (400 MHz, DMSO-*d*_6_, ppm) δ = 6.83 (s, 1H, NH), 6.69–6.63 (m, 2H, PC-H), 6.59–6.54 (m, 3H, PC-H), 6.41 (d, *J* = 8.1, 1.4 Hz, 1H, PC-H), 6.31 (d, *J* = 7.9 Hz, 1H, PC-H), 4.03–3.82 (m, 2H, ethyl-CH_2_), 3.14–2.89 (m, 6H, PC-CH_2_), 2.94–2.89 (m, 2H, PC-CH_2_), 1.23 (t, *J* = 7.2 Hz, 3H, ethyl-CH_3_). ^13^C-NMR (100 MHz, DMSO-*d*_6_, ppm) δ = 163.1 (oxadiazole-C2), 158.1 (oxadiazole-C5), 140.1 (PC-C-6’), 139.2 (PC-C-11’), 139.1 (PC-C-14’), 138.3 (PC-C-3’), 136.1, 134.1, 133.2, 133.0, 131.9, 131.6, 130.1 (PC-CH), 125.0 (PC-C-4’), 37.5 (ethyl-CH_2_), 34.8 (PC-CH_2_-1’), 34.7 (PC-CH_2_-10’), 34.5 (PC-CH_2_-9’), 33.8 (PC-CH_2_-2’), 14.6 (ethyl-CH_3_). IR (ATR, cm^−1^) *ṽ* = 3196 (w, NH), 2928 (s, Ar-CH), 2853 (m, aliph-CH), 1553 (s, C=N), 1435 (s, Ar-C=C), 1034 (s, C-O-C). MS (FAB) *m/z* (%) = 320.2 [M + H]^+^ (95). HRMS (FAB, [M + H]^+^, C_20_H_22_O_1_N_3_) calc.: 320.1763, found: 320.1762.

*5-(1,4(1,4)-Dibenzenacyclohexaphane-12-yl)-N-cyclopropyl-1,3,4-oxadiazol-2-amine* (**13e**). Violet crystals (Acetonitrile), 210 mg (63%), m.p. 222–4 °C, *R_f_* = 0.3 (Hexane: Ethyl acetate; 5:1). ^1^H-NMR (400 MHz, Methanol-*d*_4_, ppm) δ = 6.91 (s, 1H, NH), 6.67–6.59 (m, 2H, PC-H), 6.58–6.51 (m, 3H, PC-H), 6.45–6.35 (m, 2H, PC-H), 3.18–3.11 (m, 3H PC-CH_2_), 3.09–3.01 (m, 3H, PC-CH_2_), 2.98–2.91 (m, 2H, PC-CH_2_-CH_2_), 2.75–2.69 (m, 1H, cyclopropyl-CH), 0.84–0.79 (m, 2H, cyclopropyl-CH_2_), 0.67–0.63 (m, 2H, cyclopropyl-CH_2_). ^13^C-NMR (100 MHz, Methanol-*d*_4_, ppm) δ = 165.6 (oxadiazole-C2), 160.9 (oxadiazole-C5), 141.9 (PC-C-6’), 140.09 (PC-C-11’), 140.8 (PC-C-14’), 140.5 (PC-C-3’), 137.5, 136.0, 134.4, 134.3, 133.3, 133.2, 131.6 (PC-CH), 126.0 (PC-C-4’), 36.3 (PC-CH_2_-1’), 36.1 (PC-CH_2_-10’), 36.0 (PC-CH_2_-9’), 35.4 (PC-CH_2_-2’), 25.2 (cyclopropyl-CH), 7.4 (cyclopropyl-2CH_2_). IR (ATR, cm^−1^) *ṽ* = 3197 (w, NH), 2927 (s, Ar-CH), 2853 (m, aliph-CH), 1555 (s, C=N), 1435 (s, Ar-C=C), 1074 (s, C-O-C). MS (FAB) *m/z* (%) = 332.2 [M + H]^+^ (100). HRMS (FAB, [M + H]^+^, C_21_H_22_O_1_N_3_) calc.: 332.1763, found: 332.1762.

### 3.7. Crystal Structure Determinations of ***1***, ***12a***, ***12d***, ***13a***, and ***13c***

The single-crystal X-ray diffraction studies were carried out on a Bruker D8 Venture diffractometer with the PhotonII detector at 123(2) K using a Cu-Kα radiation (*λ* = 1.54178 Å). Dual space methods (SHELXT) [69] were used for the structure solution and refinement was carried out using SHELXL-2014 (full-matrix least-squares on *F^2^*) [70]. Hydrogen atoms were localized by the difference electron density determination and refined using a riding model (H(N) free, except **13a**). Semi-empirical absorption corrections were applied. Due to the bad quality of the data of **13a** the data were not deposited with The Cambridge Crystallographic Data Centre.

**1**: Colorless crystals, C_17_H_18_N_2_O, *M*_r_ = 266.33, crystal size 0.16 × 0.06 × 0.02 mm, monoclinic, space group *C*2/c (No. 15), *a* = 11.8196(4) Å, *b* = 7.9087(3) Å, *c* = 28.2370(10) Å, *β* = 92.708(2)°, *V* = 2636.58(16) Å^3^, *Z* = 8, *ρ* = 1.342 Mg/m^-3^, *µ*(Cu-K_α_) = 0.67 mm^-1^, *F*(000) = 1136, *2**θ*_max_ = 144.6°, 10645 measured reflections (2589 independent reflection in the HKLF 5 file, *R*_int_ = 0.000), 191 parameters, three restraints, *R*_1_ = 0.071 (for 2452 I > 2σ(I)), w*R*_2_ = 0.174 (all data), *S* = 1.16, largest diff. peak/hole = 0.33/−0.37 e Å^-3^. Refined as a two-component twin (BASF 0.139(4)). The option TwinRotMat of the program package PLATON [71] was used to create a HKLF 5 file, which was used for the refinement. Therefore, only unique reflections were used for the refinement (Rint = 0.00) (see cif-file for details).

**12a**: Colorless crystals, C_24_H_21_N_3_S, *M*_r_ = 383.50, crystal size 0.24 × 0.04 × 0.02 mm, orthorhombic, space group *P*ccn (No. 56), *a* = 19.8459(4) Å, *b* = 25.4981(5) Å, *c* = 7.5772(2) Å, *V* = 3834.31(15)) Å^3^, *Z* = 8, *ρ* = 1.329 Mg/m^-3^, *µ*(Cu-K_α_) = 1.60 mm^-1^, *F*(000) = 1616, *2**θ*_max_ = 144.2°, 28166 reflections, of which 3777 were independent (*R*_int_ = 0.039), 256 parameters, one restraint, *R*_1_ = 0.040 (for 3376 I > 2σ(I)), w*R*_2_ = 0.106 (all data), *S* = 1.04, largest diff. peak/hole = 0.46/−0.36 e Å^−3^.

**12d**: Colorless crystals, C_21_H_21_N_3_S·C_2_H_6_OS, *M*_r_ = 425.59, crystal size 0.24 × 0.06 × 0.02 mm, monoclinic, space group *P*2_1_/c (No. 14), *a* = 24.8195(8) Å, *b* = 7.6344(2) Å, *c* = 11.6051 (4) Å, *β* = 101.468(1)°, *V* = 2155.06(12) Å^3^, *Z* = 4, *ρ* = 1.312 Mg/m^-3^, *µ*(Cu-K_α_) = 2.38 mm^−1^, *F*(000) = 904, *2**θ*_max_ = 144.6°, 28028 reflections, of which 4256 were independent (*R*_int_ = 0.030), 267 parameters, one restraint, *R*_1_ = 0.050 (for 4009 I > 2σ(I)), w*R*_2_ = 0.134(all data), *S* = 1.07, largest diff. peak/hole = 0.89/−0.63 e Å^-3^.

**13a**: Yellow crystals, C_24_H_21_N_3_O, *M*_r_ = 367.44, crystal size 0.20 × 0.12 × 0.03 mm, monoclinic, space group *P*2_1_/c (No. 14), *a* = 13.0346(7) Å, *b* = 14.2304(8) Å, *c* = 10.0713(6) Å, *β* = 94.353(3)°, *V* = 1862.71(18) Å^3^, *Z* = 4, *ρ* = 1.310 Mg/m^-3^, *µ*(Cu-K_α_) = 0.64 mm^−1^, *F*(000) = 776, *2**θ*_max_ = 144.4°, 16984 reflections, of which 3674 were independent (*R*_int_ = 0.032).

**13c**: Violet crystals, C_20_H_21_N_3_O, *M*_r_ = 319.40, crystal size 0.16 × 0.08 × 0.02 mm, monoclinic, space group *P*2_1_/c (No. 14), *a* = 17.2016(7) Å, *b* = 8.9605(4) Å, *c* = 10.6470(4) Å, *β* = 104.112(2)°, *V* = 1591.55 (11)Å^3^, *Z* = 4, *ρ* = 1.333 Mg/m^-3^, *µ*(Cu-K_α_) = 0.66 mm^-1^, *F*(000) = 680, *2**θ*_max_ = 145.4°, 26077 measured reflections (3119 independent reflection in the HKLF 5 file, *R*_int_ = 0.000), 221 parameters, one restraint *R*_1_ = 0.066 (for 2906 I > 2σ(I)), w*R*_2_ = 0.167 (all data), *S* = 1.17, largest diff. peak/hole = 0.29/−0.32 e Å^−3^. Refined as a two-component twin (BASF 0.194(5)). The option TwinRotMat of the program package PLATON [71] was used to create a HKLF 5 file, which was used for the refinement. Therefore, only unique reflections were used for the refinement (Rint = 0.00) (see cif-file for details).

CCDC 1971268 (**1**), 1998187 (**12a**), 1998188 (**12d**), and 1998189 (**13c**) contain the supplementary crystallographic data for this paper. These data can be obtained free of charge from The Cambridge Crystallographic Data Centre via www.ccdc.cam.ac.uk/data_request/cif. Due to the bad quality of the data of 13a, the data were not deposited with The Cambridge Crystallographic Data Centre.

## 4. Conclusions

In this paper, enantiomerically pure *Sp*-*Sp*-*N*-([2.2]paracyclophanylcarbamoyl)-4-([2.2]paracyclophanylamide was synthesized from of 4-formyl-[2.2]paracyclophane (60% ee) and separated by preparative HPLC with chiral columns. We also synthesized two different classes of paracyclophanyl-heterocycles; named as 4’-[2.2]paracyclophanyl)-2,4-dihydro-3*H*-1,2,4-triazol-3-thiones and 2-amino-5-(4-[2.2]paracyclophanyl)-1,3,4-oxadiazoles. We would extend that work to include various classes of heterocyclic-paracyclophane derivatives, aiming to investigate the prospective biological and/or optical activity of these compounds.

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
