# Peer review of "Synthesis of New Planar-Chiral Linked [2.2]Paracyclophanes-N-([2.2]-Paracyclophanylcarbamoyl)-4-([2.2]Paracyclophanylcarboxamide, [2.2]Paracyclophanyl-Substituted Triazolthiones and -Substituted Oxadiazoles"

_molecules, 2020, doi:10.3390/molecules25153315_

Round 1

Reviewer 1 Report

There is a lack of explanation of the properties expected from compounds in which [2.2]cyclophane and A are linked. It cannot fully feel the importance of this research at this stage. Incorrect references were found: Refs 67, 68, and 69, although the authors described that the synthesis of compounds 2, 5, 6, 7, 8, 9, and 10 were synthesized by using the procedures written in the refs. I recommend that it is necessary to reconsider these thoroughly before accepting this manuscript.

Minor points:

In line 101,  “N-[(2.2]-para…” should be “N-[(2.2]para…”.

It must correct some garbled characters. Something has been replaced by the swirl symbol.

In line 151: It should define the abbreviation “PC” because it is the first time in this manuscript.

In line 153, it would confuse that the same compound is described as "Sp-Sp-3" and "pure chiral 3".

Author Response

There is a lack of explanation of the properties expected from compounds in which [2.2]cyclophane and A are linked. It cannot fully feel the importance of this research at this stage.

We added some sentences indicated the importance of linking chiral PC with acylhydrazide group as lines 87-93 which are:

Prompted by the aforesaid about acylhydrazide linkers and their biological activity, in addition to that planar chiral [2.2]paracyclophanes are useful synthons, from a material perspective, can be incorporated into conjugated polymeric systems for chiroptical and optoelectronic properties. These compounds show broad applications in bio- and materials science, therefore we decided to investigate the synthesis of homochiral linked paracyclophanes such as N-5-(1,4(1,4)-dibenzenacyclohexaphane-12-yl)carbamoyl)-5’-(1,4(1,4)-dibenzenacyclohexaphane-12-yl)carboxamide

Another paragraph was added to declare the importance of biological activities  of PC-linked heterocycles as:

In the other site, we previously reported that some paracyclophane-heterocycles such as methyl 2-(2-(4’-[2.2]paracyclophanyl)-hydrazinylidene)-3-substituted-4-oxothiazolidin-5-ylidene)acetates displayed anticancer activity against a leukemia subpanel, namely, RPMI-8226 and SR cell lines. The cytotoxic effect showed selectivity ratios ranging between 0.63 and 1.28 and between 0.58 and 5.89 at the GI50 and total growth inhibition (TGI) levels, respectively [62]. Therefore, we are aiming to prepare other classes of such as triazolethione and oxadiazole moieties linked to paracyclophane molecule.

Incorrect references were found: Refs 67, 68, and 69, although the authors described that the synthesis of compounds 2, 5, 6, 7, 8, 9, and 10 were synthesized by using the procedures written in the refs. I recommend that it is necessary to reconsider these thoroughly before accepting this manuscript.

Answers: Refs 69 and 70 were changed into ref 68. Ref 62; the page was appeared. refs for compounds 2-10 were corrected to be 67 and 68

Minor points:

In line 101,  “N-[(2.2]-para…” should be “N-[(2.2]para…”.

Answer: was changed into its IUPAC name. All compounds 1-3 were renamed according to IUPAC name

It must correct some garbled characters. Something has been replaced by the swirl symbol.

Were changed by symbol fonts

In line 151: It should define the abbreviation “PC” because it is the first time in this manuscript.

Answer: Was changed into paracyclophanyl

In line 153, it would confuse that the same compound is described as "Sp-Sp-3" and "pure chiral 3".

Was changed as: The structure of the compounds diastreomer-3 and/or pure chiral Sp-Sp-3 was proved by NMR spectroscopic analysis. It is clear apparent that the same number of protons and carbon signals, was assigned for both aforesaid compounds. The twenty four paracyclophanyl (PC)-aromatic carbons and eight PC-CH2-bridged carbons in addition to the two carbonyl carbons were appeared in the 13C NMR spectrum of pure chiral Sp-Sp-3, since the two paracyclophanyl moieties were electronically different via attachment with two different functionally groups

Best regards

Prof Ashraf A Aly

Reviewer 2 Report

Aly et al have presented the synthesis and characterisation of a number of interesting paracyclophane compounds.  The compounds prepared have been well characterised and the discussion of this work is clear.  I would recommend this article be accepted to Molecules with minor edits.

Line 24-25: The sentence "...we oxidized the entriosleectivity 60% ee..." it is not entirely clear what is discussed.  I presume this refers to the oxidation of compound nine which has an ee of 60%.  I would recommend adjusting the text for clarity.

Line 28: "and utilized by HPLC purification," should this "and purified by HPLC,"?

Line 43: The discussed arrangement of atoms can be known as a hydrazide linker.  Compounds with this moiety would contain a hydrazide linker rather than being considered a hydrazide linker as the sentence is currently written.

Line 110, 213 and 214: a spiral icon seems to have replaced some other symbols.

Line 135-136: What is the significance of the solvent mixture?  Is this simply an effect of improved solubility?

Line 137: the reference to the ee is slightly confusing.

Author Response

Aly et al have presented the synthesis and characterisation of a number of interesting paracyclophane compounds.  The compounds prepared have been well characterised and the discussion of this work is clear.  I would recommend this article be accepted to Molecules with minor edits.

Line 24-25: The sentence "...we oxidized the entriosleectivity 60% ee..." it is not entirely clear what is discussed.  I presume this refers to the oxidation of compound nine which has an ee of 60%.  I would recommend adjusting the text for clarity.

Answer: To prepare the homochiral linked paracyclophane of compound, the enantioselectivity of 5- (1,4(1,4)-dibenzenacyclohexaphane-12-yl)carbaldehyde (chiral purity 60% ee), was oxidized to the corresponding acid, which on chlorination, gave...........

Line 28: "and utilized by HPLC purification," should this "and purified by HPLC,"?

Answer: was changed

Line 43: The discussed arrangement of atoms can be known as a hydrazide linker.  Compounds with this moiety would contain a hydrazide linker rather than being considered a hydrazide linker as the sentence is currently written.

Answer: was corrected. As -NH-NH-CO is acylhydrazide linker not as hydrazide linker.

Line 110, 213 and 214: a spiral icon seems to have replaced some other symbols.

Answer: in line 110 was corrected. In 210 was also corrected. In 218 and 219 were also changed using symbol fonts

Line 135-136: What is the significance of the solvent mixture?  Is this simply an effect of improved solubility? 

Answer: was declared

Line 137: the reference to the ee is slightly confusing.

Answer: Reviewer has a right; ref 69 for that was changed into 68

Best regards

Prof Ashraf A Aly

Reviewer 3 Report

The authors developed a facile method to prepare a new type of planar-chiral paracyclophane. All the structures are well characterized with NMR spectroscopy and crystal structures. Overall, the manuscript is well written and ready to be published. Also, it would be interesting to see other paracyclophane derivatives synthesized by this protocol in the future.

Author Response

The authors developed a facile method to prepare a new type of planar-chiral paracyclophane. All the structures are well characterized with NMR spectroscopy and crystal structures. Overall, the manuscript is well written and ready to be published. Also, it would be interesting to see other paracyclophane derivatives synthesized by this protocol in the future.

Answer: 

I would like to thank the reviewer. There are no points of contradict or required changing.

Thanks

Best regards

Prof Ashraf A Aly

Round 2

Reviewer 1 Report

The authors have properly modified the points indicated. I recommend that this version would be acceptable to the journal.